

# Factors affecting sustainable adoption of e-health technology in developing countries: an exploratory survey of Nigerian hospitals from the perspective of healthcare professionals

Musa Ahmed Zayyad and Mehmet Toycan

Department of Management Information Systems, School of Applied Sciences, Cyprus International University, Nicosia, Cyprus

## ABSTRACT

**Background**. E-health technology applications are essential tools of modern information technology that improve quality of healthcare delivery in hospitals of both developed and developing countries. However, despite its positive benefits, studies indicate that the rate of the e-health adoption in some developing countries is either low or underutilized. This is due in part, to barriers such as resistance from healthcare professionals, poor infrastructure, and low technical expertise among others.

**Objective**. The aim of this study is to investigate, identify and analyze the underlying factors that affect healthcare professionals decision to adopt and use e-health technology applications in developing countries, with particular reference to hospitals in Nigeria.

**Methods**. The study used a cross sectional approach in the form of a close-ended questionnaire to collect quantitative data from a sample of 465 healthcare professionals randomly selected from 15 hospitals in Nigeria. We used the modified Technology Acceptance Model (TAM) as the dependent variable and external factors as independent variables. The collected data was then analyzed using SPSS statistical analysis such as frequency test, reliability analysis, and correlation coefficient analysis.

**Results**. The results obtained, which correspond with findings from other researches published, indicate that perceived usefulness, belief, willingness, as well as attitude of healthcare professionals have significant influence on their intention to adopt and use the e-health technology applications. Other strategic factors identified include low literacy level and experience in using the e-health technology applications, lack of motivation, poor organizational and management policies.

**Conclusion**. The study contributes to the literature by pinpointing significant areas where findings can positively affect, or be found useful by, healthcare policy decision makers in Nigeria and other developing countries. This can help them understand their areas of priorities and weaknesses when planning for e-health technology adoption and implementation.

Corresponding author
Musa Ahmed Zayyad,
20131260@student.ciu.edu.tr

## INTRODUCTION

### Background of the study

Over the past few decades, the field of e-health technology has witnessed significant development worldwide (*Luna et al., 2014*). E-health, as defined by the World Health Organization (WHO) is "the cost-effective and secure use of Information and Communications Technologies (ICT) in support of health and health-related fields, including healthcare services, health surveillance, health literature, and health education, knowledge and research" (*Blaya, Fraser & Holt, 2010*). E-health technology applications such as Hospital Information Systems (HIS), Electronic Medical Records (EMR) system, internet-based telemedicine, and m-health are essential tools of information technology that improve quality of healthcare delivery, increase patient safety, and reduce healthcare costs (*Shekelle, Morton & Keeler, 2006*). Other strategic benefits of e-health technology include access to up-to-date health related information, e-prescription, clinical decision support systems, well-organized information sharing among different departments, enhanced administrative system, and easy maintenance of hospital services (*Ludwick & Doucette, 2009*; *Meier, Fitzgerald & Smith, 2013*).

In most developed countries of the world, there is enormous investment of resources into acquiring the latest e-health tools and applications as a way of providing the most effective and efficient healthcare services for their citizens. These healthcare services include sharing of health information with relative ease, improvement of interaction between healthcare professionals and their patients, making access to the best healthcare services and expertise to poor and remote rural communities of a country (*Marques et al., 2011*).

In Germany for example, the government, through the Federal Ministry of Health developed electronic Health Cards for citizens covered by insurance. The smart card contains users' personal information, history of medical records, and insurance details. The card is used by patients to access healthcare services that are covered by the insurance, which significantly eases interaction between healthcare professionals and patients (*Nzuki & Mugo, 2014*). In Canada, the federal government created an independent organization called Canada Health Infoway, which is fully funded by the government and managed by Deputy Ministers of Health. This organization is charged with the responsibility of creating and promoting the use of electronic health records (EHRs) and electronic health information systems (eHIS). This ensures the sharing of medical records and health knowledge among the federal, provincial and territorial areas across the country (*Protti, 2008*).

According to *Eason & Waterson (2013)*, the United Kingdom has a well-established technology infrastructures that support e-health system. Healthcare providers have computers with internet connectivity, which they use during interaction with patients. Also, there is widespread use of Decision Support Systems (DSS) by physicians to assist in taking clinical decisions for patient's healthcare. High network connection also enables the sharing of medical data among healthcare professionals through the use of electronic health records systems. Other advanced health services in the UK includes ePrescription service, eRadiology solutions, and appointment scheduling solutions.

These strategic benefits of e-health technology make it significantly relevant for developing countries, where access to basic social amenities such as quality healthcare is hampered by poor government policies, political unrests, and lack of modern technology infrastructure (*Luna et al., 2014*). A comparative analysis of four developing countries namely Turkey, Saudi Arabia, Egypt, and United Arab Emirates was conducted by *Uluc & Ferman (2016)* to determine the challenges faced by healthcare professionals in using e-health technology. The study identified ICT infrastrucuture, policy regulations, clinical adaptation of users, healthcare financing, and supply chain management as the major challenges faced by the healthcare professionals.

However, despite the challenges faced by developing countries, studies have shown that a few have adopted and implemented a wide range of e-health tools and applications in health institutions (*Blaya, Fraser & Holt, 2010*). Studies conducted by *Verbeke, Karara & Nyssen (2013)* which evaluate the impact of ICT tools on healthcare delivery in sub-Saharan Africa have revealed the extent to which the said tools can help to improve efficiency and effectiveness of health services in hospitals. The result disclosed that clinical services such as patient identification, structured reporting, and financial management improved significantly after implementation of e-health tools in a set of 19 African health institutions. The study found also, that average waiting time decreased in 15 out of the 19 health centers, and real-time financial management metrics helped hospitals to quickly identify fraudulent practices and defective invoicing procedures. Similarly, a study conducted by *Qureshi (2016)* on creating a better world with ICT, found that technology applications such as GPS, data analysis, and cell phone records were used to track suspected cases of Ebola virus epidemic in Nigeria in 2014. So too, were the social media and SMS also used to create awareness and inform the public about the danger of the deadly disease.

Another study conducted by *Burney, Mahmood & Abbas (2010)* examined the emerging technologies in e-health in the context of developing countries. The findings showed that e-health tools such as telemedicine, m-health, bar code technology, radio frequency identification, and clinical decision support systems, picture archiving & communication system significantly improve patient safety, dietary management, and document management, which generally improve the quality of healthcare services delivered to the patients. Another study in a similar vein was conducted by *Malaquias, de Oliveira Malaquias & Hwang (2017)* on the role of ICT for development in Brazil, with specific focus on IT advances in the health area. The findings showed that e-health applications such as telemedicine applications for cancer and prescription processes, electrocardiogram in the clouds, and remote tele-monitoring of chronic patients have direct effect on social and human development.

## Problem statement

Despite the potentials of e-health solutions to improve quality healthcare in developing countries like Nigeria, studies indicate that the rate of their adoption in developing countries is either low or underutilized. This is due in part, to perceived barriers such as resistance by healthcare professionals, poor infrastructure, and low technical expertise (*Simon et al., 2007*). According to *Sapirie (2000)*, there exists a wide gap between planning for adopting

new technology, and sustainable implementation of such technology to achieve strategic or expected benefits. In order to adopt a new technology (such as e-health), there is need to conduct readiness assessment of health institutions by healthcare workers and managers to provide guidelines capable of addressing potential challenges after implementation. This assessment should be conducted both at an early stage of development and intermittently after implementation in order to evaluate the successes and challenges of the system (*Kgasi & Kalema, 2014*). Nigeria, like most developing countries, is yet to cover this gap between planning to adopt e-health technology applications, and their sustainable implementation as policy objective (*Justice, 2012*).

Nigeria had a population of 186 million in 2016 and the majority reside in remote rural and poor areas, where access to basic social amenities such as quality healthcare services, good roads, electricity supply, etc., is either poor or non-existent (*Erondu & Oladejo, 2015*). According to statistics by the Nigeria Medical Association (NMA), there are over 45,000 medical doctors in Nigeria. This indicates a ratio of one doctor to 4,000 patients (*NMA, 2016*). Clearly, it represents a far cry from the recommendations of the World Health Organization (WHO) of one doctor to not more than 600 patients. In Nigeria, the Federal government has made efforts to develop and deploy e-health technology applications in hospitals to improve healthcare services. However, healthcare workers reported that they were not carried along in the planning process, and implementation is largely at pilot stages, uncoordinated, and yet to be scaled up due to lack of comprehensive e-health national policies and strategies (*Luna et al., 2014*). Other barriers to acceptance of e-health by the healthcare professionals include apathy in embracing ICT tools, and poor awareness of e-health advantages (*Adebayo & Ofoegbu, 2014*).

## Study objective

This study specifically focuses on the factors affecting e-health technology adoption and utilization from the viewpoint of healthcare professionals in Nigeria. The objective of the study is to identify, investigate, and analyse the drivers and main barriers for e-health technology acceptance by healthcare professionals. These professionals are the primary users of the system; hence, understanding their willingness, beliefs, attitudes, and behaviours towards accepting and using the new technology is an important area to investigate and study. The findings of the study can assist healthcare administrators and policy makers in Nigeria to become better informed about e-health technology benefits, and how it can be utilized effectively to derive the maximum benefits. Moreover, by comparing the findings of the study with the existing literature, it can serve as a guide to the stakeholders to understand their areas of priorities and weaknesses when planning for e-health technology adoption and implementation in hospitals and health institutions.

The research questions for the study are:
  a.  What is the level of adoption and utilization of ICT tools in Nigerian hospitals?
  b.  What is the level of ICT literacy and experience of the healthcare professionals?
  c.  What are the challenges affecting the adoption and utilization of the ICT tools?
  d.  How can a developing country like Nigeria adopt and utilize ICT tools in hospitals?

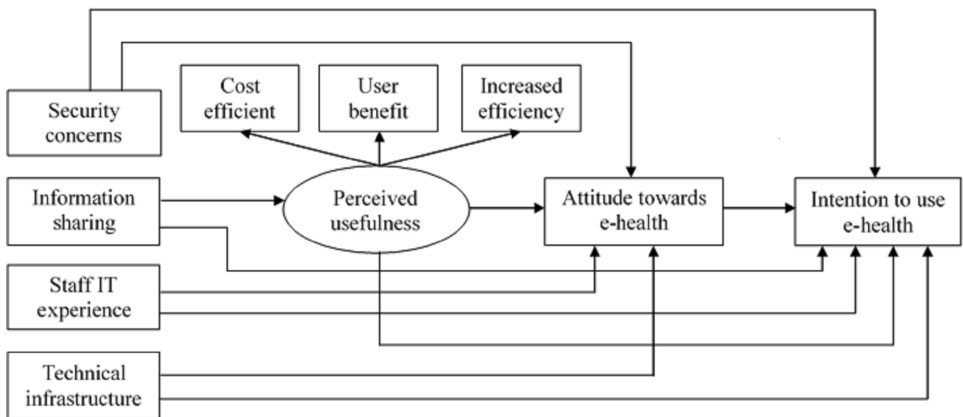

**Figure 1  Proposed theoretical research model.**

  e.  Is there correlation between the independent variables and the dependent variable of
      intention to use e-health technology by healthcare professionals in the country?

The modified TAM model (see Fig. 1) shows the relationship between the independent variables of security concerns, information sharing, staff IT experience, technical infrastructure, perceived usefulness, and attitude towards e-Health, with the dependent variable of intention to use e-Health. These variables form the basis for developing the research hypotheses that serve as the principal guides in conducting the research in order to answer the research question "e". The research hypotheses are:

*Hypothesis (H1): Attitude towards e-health will positively influence intention to use e-health.*

*Hypothesis (H2): Perceived usefulness will positively influence intention to use e-health.*

*Hypothesis (H3): Information sharing will positively influence intention to use e-health.*

*Hypothesis (H4): Staff IT experience will positively influence intention to use e-health.*

*Hypothesis (H5): Technical infrastructures will positively influence intention to use e-health.*

*Hypothesis (H6): Security concerns will positively influence intention to use e-health.*

## Theoretical model

Technology Acceptance Model (TAM) is a popular tool that is used to understand the factors affecting the adoption and usage of new technology by healthcare professionals (*Yarbrough & Smith, 2007*). This model focuses exclusively on factors determining users' behavioral intentions from healthcare professionals' perspectives. However, the TAM model is deficient in terms of integrating user interaction and task, which is an important aspect of ICT integration in organizations (*Durodolu, 2016*). Therefore, an extended TAM model is proposed and utilized in this study to identify the relative factors that contribute to the healthcare professionals' acceptance of e-health technology. Our research introduces new variables to the TAM model in order to suit this particular study. These new variables include staff IT experience, technical infrastructures, security concerns, and

information sharing. These additional independent factors enhance the TAM's predictive power (*Steininger & Stiglbauer, 2015*). Figure 1 shows the proposed research model.

The proposed theoretical model is a simple flow chart depicting the relationship between the dependent variable of intention to use e-health technology, with the independent variables of perceived usefulness, attitude towards e-health, security concerns, information sharing, staff IT experience and technical infrastructure. The theoretical model would serve as the principal guide to this study.

## MATERIALS AND METHODS

### Study design, setting and participants

This study is a cross sectional study design that was used to investigate, identify and analyze the underlying factors that affect healthcare professionals' decision to adopt and use e-health technology applications in developing countries, with particular reference to hospitals in Nigeria. The setting involves 15 hospitals in Northwest region of Nigeria. A cluster random sampling method was applied for hospital selection (*Teddlie & Yu, 2007*), while the participants were selected randomly from healthcare professionals such as doctors, nurses, radiologists, laboratory technologists, and medical directors of the hospitals. The participants were informed that participation was voluntary, and their responses would be treated as confidential. Therefore, each individual consented to participate in the survey. Previous literature studied showed that only medical doctors were considered in this type of survey, without according due consideration to other healthcare professionals. This results in a kind of bias in the research findings. Therefore, this study takes a step further by including other healthcare professionals in the survey. This ensures a balanced opinion of the participants as well as the generalizability of the research findings.

### Data collection tools

The study uses quantitative survey method in the form of a close-ended questionnaire to collect data from 15 different hospitals in the Northwest region of Nigeria. The questionnaire items (see Appendix) were adopted from a number of studies (*Tome et al., 2014*; *Erasmus, Rothmann & Van Eeden, 2015*; *Melas et al., 2011*; *Abu-Dalbouh, 2013*). These consist of demographic and survey questions asking the participants to express their opinions on their knowledge, perception, and experiences with e-health technology applications. The survey questions were rated on a 7-point Likert scale, where 1 = strongly disagree, 2 = disagree, 3 = slightly disagree, 4 = neutral, 5 = slightly agree, 6 = agree, and 7 = strongly agree. While the participants' level of Information Technology (IT) literacy and experience is measured at a scale of "None = no IT literacy"; "Minimum = little IT literacy"; "Fairly = average IT literacy"; and "Maximum = sufficient IT literacy". A total of 600 questionnaires containing 24 items were distributed, out of which 481 copies were successfully retrieved, achieving a response rate of 80.2%. However, 16 questionnaires were considered invalid due to incomplete or blank response, and thus were discarded. Therefore, only 465 surveys were included for further analysis using SPSS software version 21.

**Table 1  Demographic analysis for the survey data ($n = 465$).**

| Variable | Category | Frequency ($n$) | Percentage (%) |
| --- | --- | --- | --- |
| Gender | Male | 302 | 64.9 |
|  | Female | 163 | 35.1 |
| Profession of respondents | Doctors | 44 | 9.5 |
|  | Nurses | 144 | 31.0 |
|  | Others | 277 | 59.5 |
| Experience of respondents in hospital | 1–5 | 105 | 22.6 |
|  | 6–10 | 240 | 51.6 |
|  | 11–25 | 120 | 25.8 |

## Survey procedure

Hard copies of the questionnaires were administered by the author in person. A covering letter was attached to the questionnaire that highlights the objectives and importance of the survey. Before administering the questionnaire, the author in collaboration with the Medical Director of each hospital made a brief introduction to the participants explaining the importance of e-health and the survey in general. In order to actualize the validity of the measuring instrument used for the data collection, the questionnaire was cross-checked by academics in the field of Management Information Systems (MIS) and senior personnel in the health sector. These experts provided corrections and suggestions that result in some modifications of the items in the questionnaire, which ensure its content and face validity.

## Data analysis and management

The 465 completed surveys were then sorted, and the data was entered into SPSS software for further analysis. Each question was analyzed using a coding system that summarized the responses into subjects. Statistical tests were conducted to analyze the data, which include, reliability analysis to determine the Cronbach's Alpha value of the questionnaire, frequency test for the demographic questions, and descriptive analysis for the survey questions. The results of the demographic information and survey questions were then interpreted and analyzed in order to determine the objectives of the study, proffer solutions, recommendations, and possible direction for future research.

## Ethics

Ethical approval was obtained from the Health Research Ethics Committee (HREC) of the Federal Republic of Nigeria, with HREC clearance number: [MOH/ADM/SUB/1167/II/69].

## RESULTS AND ANALYSIS

### Demographic analysis

The result of the demographic information that includes variables such as gender, profession of participants, and their working experience are shown on Table 1.

The result shows that majority of the participants are male (64.9%), while females constitute 35.1%. This is an expected characteristic that is common in developing countries,

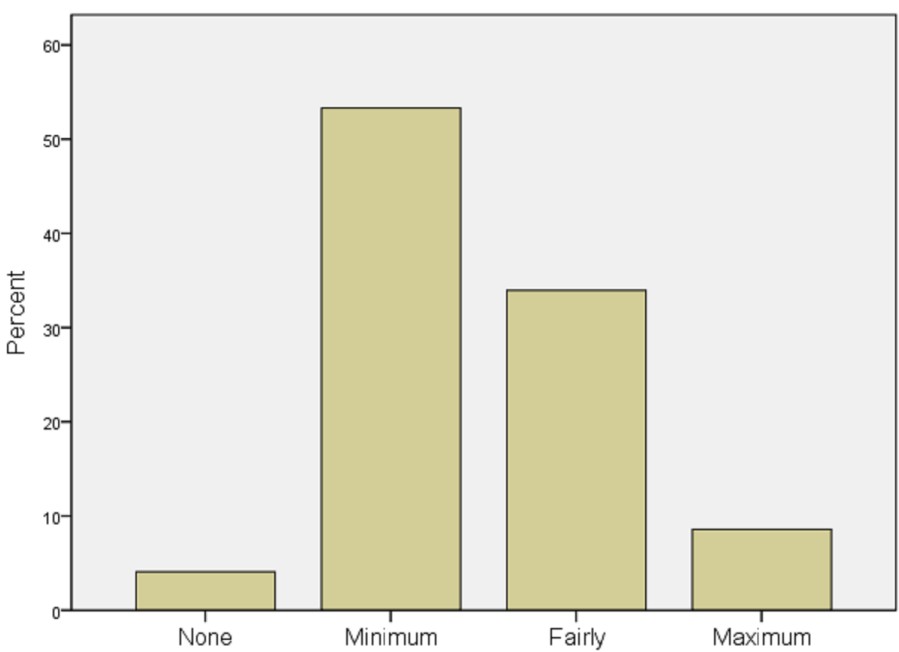

**Figure 2** **IT literacy and experience of respondents.**

where there is gender inequality in employment opportunities. According to findings by *Dormekpor (2015)*, there is gender inequality in employment opportunities, especially in developing countries. However, this did not affect the outcome of the research. The participants' working experience in terms of distribution ranged from those having 6–10 years on the job (51.6%). According to findings by *Simon et al. (2007)* and *Or & Karsh (2009)*, users of new technology in organizations tend to be younger than the average population, and have potentially higher IT literacy.

A simple descriptive analysis was conducted to determine the participants' level of IT literacy and experience, which is shown in Fig. 2.

Majority of the participants representing 53% reported having minimum level of IT literacy and experience. Similarly, 34% have fair knowledge of IT, which implies being able to use the computer, browse the internet, and send emails. About 8% reported having sufficient IT knowledge, while only 4% reported not having any knowledge of IT at all. This is an important measurement in adopting and implementing a new system. The result shows good level of IT literacy, but there is still need for improvement.

## Reliability analysis

A reliability analysis was conducted to determine the Cronbach's Alpha value of the Likert scale, which determines the consistency and reliability of the questionnaire. Reliability means that scores from the instrument are stable and consistent. The value of the Cronbach's Alpha was calculated as ($\alpha = 0.899$), which was above the recommended threshold value of 0.7 (*Tavakol & Dennick, 2011*). This further confirms the evidence of internal reliability

**Table 2  Correlation coefficient for items of the hypotheses.**

| Variables | INT | ATT | PU | TI | IS | ITE | SC |
|---|---|---|---|---|---|---|---|
| INT | – | | | | | | |
| ATT | .34 | – | | | | | |
| PU | .38 | .36 | – | | | | |
| TI | .35 | .38 | N/A | – | | | |
| IS | 05 | N/A | N/A | N/A | – | | |
| ITE | .34 | .37 | .40 | N/A | .11 | – | |
| SC | .09 | .10 | N/A | N/A | .09 | N/A | – |

Notes.

INT, Intention; ATT, Attitude; PU, Perceived Usefulness; SC, Security Concerns; IS, Information Sharing; ITE, Information Technology Experience; TI, Technical Infrastructures; N/A, Not Applicable.

of the measuring instrument, where the result is expected to remain consistent even with different samples.

## Correlation coefficient analysis

The correlation coefficient statistical analysis was conducted to determine the degree of relationship between the dependent and independent variables of the research model.

The results of the correlation coefficients shown in Table 2 indicate significant positive correlation among the variables of the hypotheses at a moderate range of ($0.34 < r < 0.40$). This means that the dependent variable from the modified TAM model can be linearly predicted from the independent variables with a significant degree of accuracy. According to *Krehbiel (2004)*, the recommended threshold value to indicate significant correlation among the variables ranges between 0.3 and 0.9.

A statistically significant positive correlation of moderate effect was obtained between attitude towards e-health and intention to use e-health (*hypothesis 1*) at $r = 0.34$. Similarly, a significant positive correlation exists between perceived usefulness and intention to use e-health (*hypothesis 2*) at $r = 0.38$. *However, the independent variable information sharing has weak positive correlation with intention to use e-Health (hypothesis 3) at $r = 0.05$.* Therefore, the hypothesis is rejected. The variable staff IT experience has significant positive correlation with the dependent variable of intention to use e-health (*hypothesis 4*) at $r = 0.34$. The independent variable of technical infrastructures also has significant positive correlation with intention to use e-health (*hypothesis 5*) at $r = 0.35$. Finally, security concerns have weak positive correlation with intention to use e-health (*hypothesis 6*) at $r = 0.09$. Therefore, the hypothesis is also rejected.

## DISCUSSION

This study was conducted to answer five principal questions, which were outlined in the introduction section.

a. *What is the level of adoption and utilization of ICT tools in Nigerian hospitals?*
   Findings from our study identified that the level of technology adoption is at a lower ebb, and even in the hospitals that were able to acquire such systems, implementation is largely at pilot stages, uncoordinated, and yet to be scaled up,

due to lack of technology infrastructure and comprehensive e-health national policies and strategies. Consequently, there was poor utilization of the technology by the healthcare professionals. This finding corresponds to those reported in other published literature (*Lee, Ramayah & Zakaria, 2012*; *Qureshi et al., 2013*) who stated that lack of technology infrastructure is the main barrier affecting healthcare professionals' decision to adopt e-health technology applications. Similarly, *Quaglio et al. (2017)* in their study on the use of Technology-Based Interventions (TBI) for healthcare delivery identified poor infrastructure and the lack of digital literacy as the major barriers to the diffusion of TBIs.

b. *What is the level of ICT literacy and experience of the healthcare professionals?*
The result of the study indicates that majority of the participants have fair knowledge of IT and a relative experience with the technology tools and applications. This finding corresponds to those obtained by *Venkatesh, Thong & Xu (2012)*, who stated that IT literacy and experience of healthcare professionals have significant influence on their willingness, perception, attitude, and intention to use health technology applications. In a similar vein, *Kabashiki & Moneke (2014)* in his study on healthcare professionals' challenge in using ICTs to manage patients identified that healthcare professionals who have sufficient information technology literacy and experience constitute positive characteristics for the use of new technology such as e-health technology applications.

c. *What are the challenges affecting the adoption and utlization of the ICT tools?*
Based on the findings from our research and the findings from previous published literature on technology acceptance studies (*Steininger & Stiglbauer, 2015*; *Melas et al., 2011*; *Venkatesh, Thong & Xu, 2012*), we identified the major barriers to e-health adoption in Nigeria and other developing countries, which we categorized into six principal areas, these include: (1) technology infrastructure barriers related to hardware, software, and networking; (2) information Technology literacy barriers related to knowledge and experience in using ICT tools and applications; (3) monetary barriers related to budgeting and funding; (4) human resource barriers related to attitude, willingness, and belief of the healthcare professionals to use the e-health technology; (5) administrative barriers related to organizational and management policies; and (6) security barriers related to privacy and trust in using the technology. This findings corresponds to those published in other studies. For example, *Qureshi et al. (2013)*, *Hoque, Mazmum & Bao (2014)* and *Jaroslawski & Saberwal (2014)* identified some challenges affecting the adoption and utilization of ICT tools in hospitals, which include poor user acceptance, lack of infrastructure, financial constraints, lack of computer literacy, and organizational policies.

d. *How can a developing country like Nigeria adopt and utilize ICT tools in hospitals?*
Findings from our study identified that there is low funding for the health sector in Nigeria. Therefore, it needs to be improved in line with the recommendations of the World Health Organization benchmark, which requires countries to allocate at least 13 percent of their annual budget to the health sector. However, Nigeria budgeted only 4.4 percent in 2016 and 4.1 percent in the 2017 national budget for the health sector, which is grossly inadequate when compared with other developing countries. This finding

corresponds to the findings of *Obansa & Orimisan (2013)*, who stated that Nigeria has a very low par capita health spending for healthcare delivery. Similarly, this findings also corresponds with those reported by *Eneji, Juliana & Onabe (2013)*, who stated that allocation of limited resources to healthcare expenditure is the major challenge facing policy makers, which in turn affects the health status and national productivity of the country. Therefore, the study identified that there is low budget and funding for the healthcare sector in Nigeria, which needs to be improved as reported by other studies.

e. *Is there correlation between the independent variables and the dependent variable of intention to use e-health technology by healthcare professionals in the country?*
The results obtained through correlation coefficient statistical analysis for hypothesis testing to determine the degree of relationship among the variables indicated the significant positive correlation between the dependent variable of intention to use e-health and the independent variables of the research hypothesis (see Table 2). These findings corresponds with results from other published studies (*Adebayo & Ofoegbu, 2014*; *Bergoeing, Loayza & Piguillem, 2010*; *Hoque, Mazmum & Bao, 2014*; *Qureshi et al., 2013*), who stated that technical infrastructures, IT literacy and experience, as well as attitude, belief, and willingness of the healthcare professionals have significant influence on their intention to use the e-health technology applications.

## Limitations and future research

Our study is limited by geographical scaling, having been conducted in one country and focused on 15 hospitals only. This is owing mainly to shortage of funds, as grants for research are not easily accessible or readily obtainable. Therefore, there is room for making its scope wider and its findings more reliable and generalized if more countries and additional hospitals are visited and examined. However, based on the related literature reviewed, we were able to identify and study the situation in other countries. We suggest that future researchers would fill in whatever yawning gap that is perceived between our current findings and what vistas remained to be uncovered.

Another limitation is that the study is cross sectional, which is considered fast, simple, and less expensive to perform. However, a cross sectional study could be biased, where participants who took part in the study failed to represent the views of the whole population, affecting consequently, the generalizibility of the research findings. This is because it is based on a questionnaire survey, and participants are interviewed only once (*Sedgwick, 2014*). Hence, future study should consider adopting a longitudinal approach, where participants are interviewed severally over a period of time.

However, despite these limitations, this study was able to investigate and identify the critical factors that affect healthcare professionals' decision and willingness to adopt and implement e-health technology applications in developing countries with specific focus to Nigerian hospitals.

## CONCLUSION

This study focused mainly on the pertinence and necessity of adopting and implementing e-health technology applications in hospitals and medical institutions in a developing

country like Nigeria. The main objective of the research was to determine the critical barriers and drivers for adopting e-health technology applications from the viewpoint of healthcare professionals in developing countries. The findings from this study would make significant contributions both to theory and practice of sustainable information system adoption in Nigeria relating to the health sector and its effective delivery and management. The technology acceptance theory which evolved from our research could be refined for the attainment of the said objective, as much in Nigeria as in other developing countries.

## ACKNOWLEDGEMENTS

We would like to thank all those healthcare professionals who participated in filling the questionnaire, and we particularly acknowledge the contribution of Dr. Mustafa Abu-Yaro (The Chief Medical Director), who assisted in mobilizing the participants of the survey.

### Funding
The authors received no funding for this work.

### Competing Interests
The authors declare there are no competing interests.

### Author Contributions
- Musa Ahmed Zayyad performed the experiments, analyzed the data, contributed reagents/materials/analysis tools, prepared figures and/or tables, authored or reviewed drafts of the paper, approved the final draft.
- Mehmet Toycan conceived and designed the experiments, contributed reagents/materials/analysis tools, authored or reviewed drafts of the paper, approved the final draft.

### Human Ethics
The following information was supplied relating to ethical approvals (i.e., approving body and any reference numbers):

The Health Research Ethics Committee (HREC) of the Federal Republic of Nigeria granted Ethical approval to carry out the study. Approval number: MOH/ADM/-SUB/1167/II/69.

### Data Availability
The SPSS raw data used for the statistical analysis has been provided as a Supplemental File.

### Supplemental Information
Supplemental information for this article can be found online at http://dx.doi.org/10.7717/peerj.4436#supplemental-information.

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
