# Peer review of "Factors affecting sustainable adoption of e-health technology in developing countries: an exploratory survey of Nigerian hospitals from the perspective of healthcare professionals"

_PeerJ, doi:10.7717/peerj.4436_

## Round 0.1 · original submission · Major Revisions

Dear Authors,

The Reviewers found your manuscript interesting, however they raised serious concerns that, at the moment, do not allow us to consider your manuscript for publication.

I would suggest to take into consideration the Reviewers' comments, discuss and address all of them within your manuscript and a point-by-point rebuttal letter.

Best regards

Salvatore Andrea Mastrolia
PeerJ Academic Editor

Reviewer 1 ·

Basic reporting

The authors used clear and easy to understand English
Background of the study
• Review of literature on study’s background is scanty and does not present a clear picture the said topic. A more rigorous literature relating to the acceptance of e-Health by care providers would be needed
Problem statement
• The authors state from line 6 … there exist a wide gap… does this gap relate to adoption by care providers?
• On line 11 the authors have to be precise regarding the population of Nigeria (delete “about”
• Reference is needed at the end of line 15.
• I am not sure about the clarity of the problem statement – Seems to be motivation for the adoption of e-Health by healthcare organizations – acceptance at organizational level – Relate problem statement to individual acceptance of e-Health

Experimental design

Study Objectives
• Research question “e” could have been answered by formulating and testing hypotheses relating to various constructs in the model to verify any association between the identified factors and the dependent/outcome variable
Theoretical background
• I am not sure what theory underpins this research. On the one hand the authors state theory but did not make reference to any particular theory. Instead the authors chose to talk about TAM under research model. TAM, could have been placed under theory instead of research model
Research model
• On line 15 the authors introduce “patients…?
• On line 21 are human capacity and organizational factors. But these factors/constructs are not present in figure 1.
Sampling of participants
• On line 21 are “non-medical staff… But these participants are clearly not healthcare professionals/providers!
• Regarding the cover letter and the questionnaire – Did the authors assumed every healthcare provider or staff of healthcare facilities in Nigeria already know what e-health is?

Validity of the findings

Analysis
• The authors state they have introduced three constructs (Human resource, …) to make up for the deficient TAM model. The authors do not state how those constructs were measured? Lots of missteps
Discussion
• I'm afraid all the proposed questions have not been fully answered.

Reviewer 2 ·

Basic reporting

This interesting piece of work aims to investigate, identify and analyse the underlying factors that affect healthcare professionals decision to adopt and use e-health technology applications in developing countries, with particular reference to hospitals in Nigeria. That's well run and written and deserves consideration for publication. However, I have some minor comments authors may wish to address.

Experimental design

That's a cross sectional survey. Classical limitations should be considered.

Validity of the findings

That's a cross sectional survey. Classical limitations should be considered.

Additional comments

That's a cross sectional survey. Classical limitations should be considered. Please use for example http://www.bmj.com/content/348/bmj.g2276?variant=full-text&hwshib2=authn%3A1513021746%3A20171210%253Ac207388f-8e7f-4a58-afe9-4041e6889f8c%3A0%3A0%3A0%3AIlFK0SLMhD6Ywi7DihLqJQ%3D%3D for limitations and https://www.strobe-statement.org/fileadmin/Strobe/uploads/checklists/STROBE_checklist_v4_cross-sectional.pdf for reporting issues.
Also, I feel some wider International comparisons should be made using for example https://www.karger.com/Article/Abstract/478904 and https://www.ncbi.nlm.nih.gov/pubmed/27107907

---

## Round 0.2 · Major Revisions

Dear Authors,

As you can see from the comments of Reviewer 2, it is felt that you need to further expand your treatment of the international literature to correctly place the article in a wider context (regarding comparisons to other work and possible alternate explanations).

Best regards

Salvatore Andrea Mastrolia
PeerJ Academic Editor

Reviewer 2 ·

Basic reporting

Acceptable

Experimental design

Acceptable

Validity of the findings

I feel a wider use of the International literature in terms of comparisons and alternative explanations is still needed.

Additional comments

I feel a wider use of the International literature in terms of comparisons and alternative explanations is still needed.

---

## Round 0.3 · accepted · Accept

Dear Authors,

I would like to compliment with you for the efforts provided in addressing the Reviewers' comments.

I feel that your manuscript has reached the level of publication and can be accepted in its current form.

Best regards

Salvatore Andrea Mastrolia
PeerJ Academic Editor